# TAMC: Textual Alignment and Masked Consistency for Open-Vocabulary 3D Scene Understanding

**DOI:** 10.3390/s24196166

**Published:** 2024-09-24

**Authors:** Juan Wang, Zhijie Wang, Tomo Miyazaki, Yaohou Fan, Shinichiro Omachi

**Affiliations:** 1Department of Communications Engineering, Graduate School of Engineering, Tohoku University, Sendai 9808579, Japan; wang.juan.t4@dc.tohoku.ac.jp (J.W.); tomo@tohoku.ac.jp (T.M.); fan.yaohou.t4@dc.tohoku.ac.jp (Y.F.); 2RIKEN AIP, Tokyo 1030027, Japan; zhijie@vision.is.tohoku.ac.jp

**Keywords:** open vocabulary, 3D Scene Understanding, multi-modal learning, contrastive learning, Masked Consistency, Textual Alignment

## Abstract

Three-dimensional (3D) Scene Understanding achieves environmental perception by extracting and analyzing point cloud data with wide applications including virtual reality, robotics, etc. Previous methods align the 2D image feature from a pre-trained CLIP model and the 3D point cloud feature for the open vocabulary scene understanding ability. We believe that existing methods have the following two deficiencies: (1) the 3D feature extraction process ignores the challenges of real scenarios, i.e., point cloud data are very sparse and even incomplete; (2) the training stage lacks direct text supervision, leading to inconsistency with the inference stage. To address the first issue, we employ a Masked Consistency training policy. Specifically, during the alignment of 3D and 2D features, we mask some 3D features to force the model to understand the entire scene using only partial 3D features. For the second issue, we generate pseudo-text labels and align them with the 3D features during the training process. In particular, we first generate a description for each 2D image belonging to the same 3D scene and then use a summarization model to fuse these descriptions into a single description of the scene. Subsequently, we align 2D-3D features and 3D-text features simultaneously during training. Massive experiments demonstrate the effectiveness of our method, outperforming state-of-the-art approaches.

## 1. Introduction

Three-dimensional (3D) Scene Understanding is a fundamental computer vision task; its aim is to perform point-wise classification on input point cloud data. Three-dimensional Scene Understanding has various applications, including virtual reality, robot manipulation, and human–machine interaction [1,2]. Traditional fully supervised methods [3,4] rely on fully annotated data leading to time-consuming labeling and failure to recognize unseen categories. Some methods [5,6] divide datasets into seen and novel classes for training and testing, respectively—as shown in Figure 1a—to reduce the heavy annotation costs. However, these methods can still only work with fixed categories; more specifically, they need to know what the novel classes are and cannot handle queries with arbitrary text input. Open-vocabulary 3D Scene Understanding is proposed to address the above problems by enabling the model to recognize arbitrary input text, not limited to close-set dataset categories. The diversity of potential queries makes it challenging, and has prompted many studies [7,8,9] to explore these problems.

Existing state-of-the-art method OpenScene [7] leverages CLIP [10] to distill knowledge into the 3D domain via contrastive learning. CLIP is a model pre-trained on a large number of image-text pairs; OpenScene aligns the 3D features with 2D features extracted from CLIP model, establishing the connection between 3D data and text, as illustrated in Figure 1b. OpenScene does not require the use of any 3D-text data pairs for training, thus significantly reducing annotation costs.

After dissecting the previous work, we found two insufficiencies. Firstly, the 3D feature extraction process is not robust enough. Specifically, during the training of OpenScene, the features of the complete 3D point cloud are used, while in the real scenario, the point clouds of scenes are incomplete and partially missing. Secondly, OpenScene connects 3D features and text by aligning 2D and 3D features, lacking direct supervision from the text. These two limitations both constrain the performance of OpenScene. As shown in Figure 1c, to address the first problem, we used Masked Consistency (MC, refer to Section 3.2) in training, where part of the features are randomly masked during training to force the model to understand the global scene using only partial 3D features. For the second problem, we first generated text pseudo labels for the point cloud and used them to assist the training process, which is the Textual Alignment method (TA, refer to Section 3.3), allowing the text to directly supervise the training process of the 3D feature extractor.

In summary, the main contributions of this paper are three-fold:We propose Masked Consistency training, making the extracted feature more robust for real-world 3D Scene Understanding tasks.We use a generated text pseudo label to assist the training process, which is Textual Alignment, to enable better interaction between the 3D features and text features.We conduct extensive experiments with the proposed method and update the state-of-the-art performance, demonstrating the effectiveness of the proposed method.

## 2. Related Work

### 2.1. Three-Dimensional Scene Understanding

Three-dimensional Scene Understanding focuses on understanding the semantic meaning of objects and environments from point clouds, with 3D semantics being a fundamental perception task for Scene Understanding. For semantic feature extraction and prediction from point clouds, existing methods have designed custom point convolutions applied to the raw point cloud [11,12,13]. Alternatively, some methods transform point clouds into 3D grids (voxels) and employ sparse convolutions [14], voxel-tailored networks [15], or transformers [16] for feature extraction. Despite achieving an outstanding performance on closed-set benchmark datasets, they often struggle with open-world settings, i.e., the recognition of unseen categories not annotated during training.

### 2.2. Two-Dimensional (2D) Scene Understanding

Two-dimensional (2D) Scene Understanding models are designed to allow users to interact with input images through text. Pioneering work in this field is CLIP [10]. It is trained on a vast dataset of image–text pairs, enabling it to understand the relationship between visual content and linguistic descriptions. CLIP’s versatility allows it to perform various tasks, such as zero-shot image classification and cross-modal retrieval, without task-specific fine-tuning. To ensure good generalization, CLIP requires training on a large number of image–text pairs. Recently, various new methods have been proposed for augmentation, including prompt learning [17,18], module fine-tuning [3,19], and knowledge distillation [20,21]. This work enables open-world tasks like segmentation [3,19,22], detection [20,21], and more.

### 2.3. Zero-Shot Learning for 3D Point Clouds

Inspired by the success of 2D Scene Understanding models [10,23], some works have adopted similar strategies for the zero-shot learning for 3D point clouds, which aims to classify 3D points without ground truths. For instance, PLA [5] first uses a caption model to generate pseudo textual descriptions for 2D images corresponding to 3D point clouds, then aligns these descriptions with 3D features in a contrastive learning fashion. However, challenges like inaccurate predictions persist due to coarse language supervision. RegionPLC [6] is proposed to address it, which introduces a pre-trained detection model to densely detect the input 2D image, thereby obtaining richer text descriptions. However, during the training process, both PLA and RegionPLC divide the dataset into seen classes and unseen classes; the ground truth is available for the former, while the latter uses the previously mentioned methods to generate pseudo labels. Subsequently, the model achieves point-wise classification through supervised training on both seen and unseen classes. These methods still rely on supervised training with 3D ground truth labels (seen classes) or pseudo labels (unseen classes); they can only segment pre-defined classes. OpenScene [7] enables class-agnostic classification without requiring labeled 3D data. It first uses a 3D feature extractor to obtain embeddings for each 3D point, and then a frozen image encoder of the pre-trained CLIP is applied to generate 2D embeddings corresponding to each 3D point. OpenScene aligns 2D and 3D features through contrastive learning. Since 2D and text features have already been aligned during CLIP’s training process, after the 2D–3D alignment of OpenScene, 3D features also establish a connection with text, thereby enabling point-wise classification. However, OpenScene does not consider the inherent sparsity of point cloud data, and it also lacks direct supervision from textual information during the training process, resulting in suboptimal performance. The proposed method alleviates the two aforementioned problems by introducing the masking policy and pseudo-textual descriptions during training, thereby achieving better scene understanding capabilities.

## 3. Method

Open-vocabulary 3D Scene Understanding models, such as OpenScene, use images as supervision, allowing the training process to proceed without any semantic-level annotations. However, the lack of textual information during training results in sub-optimal performance. In contrast, PLA introduces a language-driven training process to achieve open-vocabulary recognition, but the requirement for predefined categories limits the model’s scalability. Therefore, the proposed method aims to combine the strengths of various modalities (such as images, point clouds, and text) to train a model that enhances robustness and performance without predefined categories. Our framework builds upon OpenScene; thus, we will first revisit OpenScene.

### 3.1. Revisiting OpenScene

As illustrated in Figure 1b, OpenScene first extracts point-wise 3D features $F^{3D}$ and 2D features $F^{2D}$ of a 3D scene from unlabeled 3D and 2D data for training, respectively. Thus, we can obtain $F^{3D}$ via:(1)$$F^{3D}=\epsilon_{\theta }^{3D}\left(P\right)$$
where $P\in R^{M\times 3}$ represents the input point clouds of a scene with *M* points, $\epsilon_{\theta }^{3D}:R^{M\times 3}\mapsto R^{M\times C}$ is a trainable 3D encoder, which adopts MinkowskiNet18A [15] to get per-point features $F^{3D}$, *C* is a feature dimension, and $F^{3D}=\left\{f_{1}^{3D},\ldots ,f_{M}^{3D}\right\}$ represent 3D features with *M* points.

For each 3D point, the corresponding 2D pixels can be calculated according to the intrinsic matrix and world-to-camera extrinsic. Then a frozen 2D encoder $\epsilon_{\theta }^{2D}$ and an average pooling operator P are employed to extract multi-view pixel-wise features f and fuse them, respectively. This process can be expressed by the following equation: $f^{2D}=P\left(f_{1},\ldots ,f_{K}\right)$, where a total number of *K* pixels can be associated with one point. $F^{2D}$ can be obtained by repeating the fusion process for each point:(2)$$F^{2D}=\left\{f_{1}^{2D},\ldots ,f_{M}^{2D}\right\}\in R^{M\times C}$$
where *M* presents the points number of the input point clouds; it means the length of $F^{2D}$ vector is consistent with $F^{3D}$. So that 2D features $F^{2D}$ can be aligned with 3D feature $F^{3D}$. It is worth noting that OpenScene utilizes the LSeg [3], a CLIP variant fine-tuned for the pixel-wise classification task. We follow the same setting for fair comparisons. The final step is to align the 2D and 3D modalities via contrastive learning between 2D features $F^{2D}$ and 3D features $F^{3D}$; the objective function is:(3)$$L=1-cos(F^{2D},F^{3D})$$
where cos means the cosine similarity calculation operation. Since the 2D features $F^{2D}$ have already been aligned with the textual features $F^{text}$ during the CLIP pre-training, aligning the 3D features $F^{3D}$ with $F^{2D}$ via Equation (Equation 3) will also enable $F^{3D}$ to be consistent with $F^{text}$. This allows the model to achieve open vocabulary scene understanding in 3D scenarios during testing, as shown in Figure 1b.

### 3.2. Three-Dimensional Feature for Masked Consistency Training

Point clouds in the real world are often sparse or even partially missing. The model needs to understand the complete scene through incomplete 3D features. OpenScene fails to consider this characteristic of 3D data, leading to sub-optimal performance. Motivated by this finding, we propose a new training policy to improve the model, named Masked Consistency (MC). Without bells and whistles, we mask some features of $F^{3D}$ randomly to get a new representation $F_{Mask}^{3D}$, which can be represented by the following equation:(4)$$F_{Mask}^{3D}=F^{3D}⊙M_{r}$$
where $M_{r}$ is a randomly generated [0, 1] mask with the same spatial size of $F^{3D}$, and the *r* is a scalar between 0 and 1, which means the ratio of 0, i.e., $M_{0.6}$ is a mask where 60% of the mask is randomly filled with zeros, and the remaining parts are filled with ones. ⊙ means the Hadamard product. The proposed MC encourages the model to predict complete scene information through partial point cloud features, making the trained model more generalized and robust. We validate the influence of different mask ratios on model performance in Section 4.3.4.

### 3.3. Text Feature for Textual Alignment Training

OpenScene aligns the features of 2D and 3D data during training. Specifically, the 3D representation uses the 2D data as an intermediary to connect with the text, thereby acquiring scene understanding capabilities. This training workflow lacks direct interaction between the 3D data and the text, leading to sub-optimal performance. We aim to address this issue and enhance the model’s effectiveness. Specifically, we first generate corresponding pseudo-text features for the 3D point clouds and then directly align the 3D point clouds with these pseudo-text features to improve performance.

We borrow the workflow from PLA [5] to generate pseudo-text descriptions for point clouds. Specifically, for a 3D scene paired with multi-view RGB images, we use the ViT-GPT2 [24], a pre-trained image captioning model, to generate textual descriptions for images:(5)$$t_{i}=G_{cap}\left(I_{i}\right)$$
where $G_{cap}$ is the pre-trained caption model, $I_{i}$ and $t_{i}$ is the *i*th 2D input image and the corresponding generated caption, respectively.

Then, a pre-trained text summarizer, BART [25], is used to summarize all captions $t_{i}$ of a scene:(6)$$t=G_{sum}\left(t_{1},\ldots ,t_{i},\ldots ,t_{N}\right)$$
where $G_{sum}$ is the pre-trained caption summarizer, t is the final scene-level caption that describes the entire scene, *N* is the number of frames in a scene.

The captioner $G_{cap}$, as shown in Figure 2, has been pre-trained on a massive number of image–text pairs, exhibiting strong generalization capabilities. As a result, it can generate rich semantic descriptions for 2D images, which include attributes of entities as well as the relationships between them. For example, in $t_{1}$, “a kitchen with a wooden table and chairs” corresponds to the first image caption. In this text description, “wooden” describes the material, “a” and “s” indicate quantity, “kitchen” specifies the room type, and “with” explains the spatial relationship. Compared to simply representing visual information with single entities like “table” and “chairs”, such a complete description provides richer semantic information and interaction relationships. Moreover, after summarizing multi-view descriptions using $G_{sum}$, we can retain effective information and avoid potential semantic conflicts that may arise in multi-view descriptions. We validate the effectiveness of scene-level text descriptions for model performance in Section 4.3.

Then we use the text encoder from a pre-trained CLIP [10], denoted as $\epsilon_{\theta }^{Text}$, to encode the final caption and get the text features $F^{Text}$ as follows:(7)$$F^{Text}=\epsilon_{\theta }^{Text}\left(t\right)\in R^{512}$$
where $\epsilon_{\theta }^{Text}$ is frozen during training and testing, as shown in Figure 3. During training (Figure 3a), we use the scene-level caption generated in Figure 2 as the input of $\epsilon_{\theta }^{Text}$. For testing (Figure 3b), we prompt the input query via the template ‘*a XX in a scene*’, and feed it into $\epsilon_{\theta }^{Text}$. For example, we use “a sofa in a scene” as the text input to identify the points of the sofa.

### 3.4. Textual Alignment and Masked Consistency via Contrastive Learning

As shown in Figure 3, after the generation of multi-modal features, including 2D image features $F^{2D}$, text features $F^{Text}$, and Masked 3D point cloud features $F_{Mask}^{3D}$, we can align them through contrastive learning. Specifically, we use the following objective functions:(8)$$L_{1}=1-cos\left(F^{2D},F_{Mask}^{3D}\right)$$
(9)$$L_{2}=1-cos\left(F^{Text},F_{Mask}^{3D}\right)$$
(10)$$L_{TAMC}=L_{1}+\alpha L_{2}$$
where cos is the cosine similarity calculation function. $L_{1}$ is to align the masked 3D features $F_{Mask}^{3D}$ and 2D features $F^{2D}$ and $L_{2}$ is to align the $F_{Mask}^{3D}$ and text features $F^{Text}$. α is a weight to balance the loss of different modalities; we discuss its influence in Section 4.3.2. Please note that $\epsilon_{\theta }^{2D}$ and $\epsilon_{\theta }^{Text}$ are frozen during the training process and only $\epsilon_{\theta }^{3D}$ is trainable.

For the multi-modalities’ alignment, due to the open vocabulary image features and text features, the output of the refined 3D model naturally exists within the same feature space. Therefore, TAMC does not require predefined categories as PLA [5] does. This is evident from the experimental setup, where we do not distinguish between base and novel classes like PLA. Any category or arbitrary input is acceptable. Moreover, this text-image joint feature in $F^{3D}$ allows for 3D scene-level understanding given any textual prompt. Compared to OpenScene, TAMC’s semantic understanding capability is superior, as validated in Section 4.2 through comparisons with the state-of-the-art methods. Additionally, the Masked Consistency strategy enhances the model’s recognition of irregular objects, where we demonstrate the result in Section 4.3.3.

## 4. Experiments

### 4.1. Setups

#### 4.1.1. Dataset

We use ScanNet [26] as the benchmark dataset, which contains indoor scene data annotated with 20 classes for the pixel-wise labeling task.

#### 4.1.2. Metrics

We calculate class-wise intersection over union (IoU) and accuracy (Acc.) and report their mean values, i.e., mean IoU (mIoU) and mean Acc. (mAcc.) for evaluation.

#### 4.1.3. Model Structure

We keep the identical setting as OpenScene for fair comparisons, where LSeg [3] is utilized; it is a CLIP variant for the image segmentation task. LSeg consists of an image encoder and a text encoder, which are used to generate image and text features, respectively. For the 3D encoder, we use MinkowskiNet18A [15].

### 4.2. Results

In Table 1, we evaluate the proposed method on the ScanNet [26] validation set and compare its performance against both fully supervised and zero-shot baselines. To ensure a fair comparison, we reproduce the results for OpenScene based on the official codes released by the authors (https://pengsongyou.github.io/openscene (accessed on 20 September 2024)). For zero-shot methods, TAMC achieves state-of-the-art performance, with mIoU and mAcc scores of 51.9% and 63.4%, respectively. These scores surpass those of the OpenScene baseline by 1.3 and 1.2 percentage points, demonstrating the effectiveness of the proposed method. During fully supervised methods, although it outperforms those from previous years [27], there is still a significant gap compared to the state-of-the-art methods [28]. This indicates that zero-shot methods have considerable potential in 3D understanding tasks.

Visual comparisons of semantic segmentations are shown in Figure 4. In the top row, the result highlighted in the red box indicates that TAMC predicts the bathtub (marked in pink) more accurately and completely. In the second row, a similar phenomenon can be observed, where our prediction for the door is more precise. In the third row, the proposed method corrects the wrong prediction of the baseline (the red door). These results suggest that by incorporating the proposed Textual Alignment and Masked Consistency by contrastive learning, we can achieve better and more robust scene understanding capabilities, leading to improved performance.

### 4.3. Ablation Studies and Analysis

#### 4.3.1. Influence of Textual Alignment Training

To investigate the impact of Textual Alignment (TA) on model performance, we conducted some comparative experiments. As shown in Table 2, the baseline (0) represents the original OpenScene, which achieves an mIoU score of 50.6%. By incorporating the 3D-text loss function $L_{1}$ and combining it with the 3D–2D loss function $L_{2}$ using a weight α (as detailed in Equation (Equation 10)), we observed a notable improvement in the model’s performance. It is important to note that, in order to exclude the influence of the proposed Masked Consistency training, we utilize the full 3D features without masking and align them with both text and image features.

#### 4.3.2. Balance Weights of Textual Alignment Training

We introduced a parameter α to balance the 3D-text loss $L_{1}$ and the 3D–2D loss $L_{2}$, which influences the model’s final performance. As illustrated in Table 2, we experimented with different α, finding that the model achieves optimal performance when α=0.05.

#### 4.3.3. Influence of Masked Consistency Training

We discuss the impact of the proposed Masked Consistency (MC) in this section, the experimental results are shown in the leftmost column of Table 3. Firstly, the MC can boost performance even with different masking ratios, except for the masking ratio of 0.1; other masking ratios can lead to an improvement of 0.2 to 1.0 in terms of the mIoU score. The reason for our improved performance lies in the proposed method’s better understanding of sparse 3D data. Specifically, MC forces the model to understand the complete scene with only a partial point cloud during the training process, enabling it to obtain richer information during inference.

#### 4.3.4. Masking Ratio Selection for Masked Consistency Training

For Masked Consistency (MC), varying masking ratios cause performance shifts, according to the results shown in Table 3. In particular, when the masking ratio is set to 0.95, MC achieves the best mIoU score of 51.6%, which is 1.0 pp. higher than the baseline model. When we set the masking ratio to 0.1, the mIoU score is 50.6%, which is on par with the baseline OpenScene. Experimental results indicate that larger masking ratios could lead to greater improvements. This is consistent with observations in previous masking training policies for images [34], where large masking ratios are often used. Please note that we do not mask 3D point clouds directly. Instead, we mask the 3D features $F^{3D}$, i.e., the output of $\epsilon_{\theta }^{3D}$ in Figure 3. $F^{3D}$ has the same spatial size as the input point cloud; this policy brings simplified engineering implementation and avoids potential misalignment during data preprocessing. It is possible that identical masking ratios would lead to different occlusions for scene features because of the proposed random masking strategy, thus introducing slight fluctuations in performance. With our masking policy, each embedding corresponds to a point that actually contains information from other points as well. Thus, a masking ratio of 0.99 does not imply that merely 1% of the raw input 3D data are available; rather, it signifies that we only present 1% of the extracted feature information. There are minor fluctuations (±0.2) in terms of mIoU score when varying the masking ratio between 0.6 and 0.95. This stability across different masking ratios highlights the robustness and adaptability of our method. This observation aligns with the findings of the MAE [34], a pre-training method designed for 2D images. When the masking ratio changes between 0.4 and 0.8, its fluctuations on the classification task are only ±0.3.

#### 4.3.5. The Training Policy

TAMC adopts a multi-stage training pipeline for optimal performance. We conducted several ablation experiments to study the order of the proposed Masked Consistency (MC) and Textual Alignment (TA). We define two training strategies: (1) MC → TA, in which we first train the model using MC and then fine-tune it with a combination of 2D–3D and text-3D alignment losses (Equation (Equation 10)); (2) TA → MC, which represents the opposite training order of that in 1). As shown in Table 3, the performance of TA → MC consistently surpasses that of MC → TA across different masking ratios. Specifically, when the masking ratio is set to 0.95, and the training strategy is TA → MC, the model achieves an mIoU score of 51.9%, while the best result for MC → TA is only 51.0%. Therefore, we choose the second training strategy as the final one.

#### 4.3.6. The Analysis of Per-Class Results

Table 4 shows the results of class-wise IoU; our method demonstrates superior performance compared to the baseline in most categories, particularly in those with irregular shapes like chairs and other furniture. It also works well on categories that may appear irregular due to overlap with other objects, such as floors, walls, cabinets, tables, and bookshelves. However, it also shows sub-optimal performance in some categories. On one hand, this might be attributed to the scarcity of training data; for example, classes like ‘picture’ and ‘toilet’ occupy a very small proportion of the overall training dataset. On the other hand, some categories of ScanNet exhibit high semantic similarity, such as ‘curtain’ and ‘shower curtain,’ which poses significant challenges for precise point-wise classification.

## 5. Conclusions

In this paper, we proposed a novel contrastive learning framework based on Textual Alignment (TA) and Masked Consistency (MC) training for the open vocabulary 3D Scene Understanding task. They are proposed to address the problems of existing methods, i.e., the lack of direct textual supervision and the neglect of the sparsity of 3D data, respectively. It masks part of the 3D features and then aligns them with 2D features and textual features through contrastive learning. The training process of TAMC incorporates direct interaction between 3D features and textual information while also forcing the model to understand the entire scene from partial 3D information. The proposed method has better scene understanding capabilities and robustness, resulting in improved performance. Our experiments demonstrate the effectiveness of the proposed method, which outperforms the baseline and achieves new state-of-the-art. The proposed method currently generates textual information for the entire scene, which is coarse for point-wise classification tasks; generating more fine-grained text could be a promising research direction for future work.

## Figures and Tables

**Figure 1 sensors-24-06166-f001:**
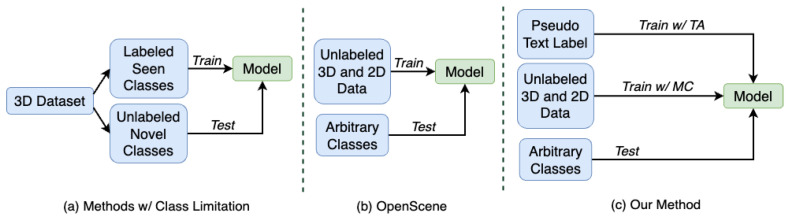
Comparison of previous methods and ours. (**a**) is the workflow of methods with class limitation, which needs labeled 3D point cloud data of seen classes and evaluates the model on unlabeled classes. As a comparison, both (**b**) OpenScene [7] and (**c**) our method follow the open vocabulary setting, meaning that no annotated data are required during the training process. The difference lies in that the proposed training process includes the supervision from the pseudo text label, and the masking training policy makes the extracted 3D features more robust, resulting in higher accuracy.

**Figure 2 sensors-24-06166-f002:**
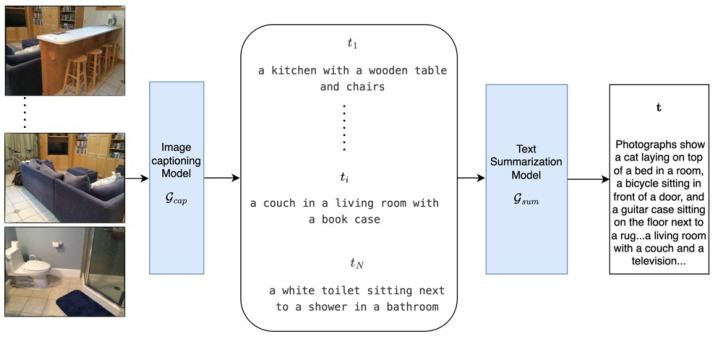
Caption generation. Multi-view images are fed into the image captioning model $G_{cap}$ to generate corresponding captions $t_{i}$ of a scene with *N* images, then the text-summarization model $G_{sum}$ summarizes $\left(t_{1},\ldots ,t_{i},\ldots ,t_{N}\right)$ to generate a scene-level caption t.

**Figure 3 sensors-24-06166-f003:**
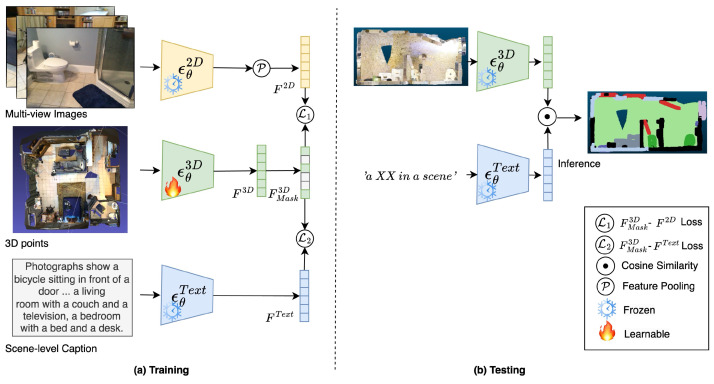
Method overview. (**a**) Training. Given a 3D point cloud, a set of posed images, and a scene-level caption, we train a 3D encoder $\epsilon_{\theta }^{3D}$ to produce a masked point-wise 3D feature $F_{Mask}^{3D}$ with two losses: $L_{1}$ and $L_{2}$ for $F_{Mask}^{3D}$-$F^{2D}$ loss and $F_{Mask}^{3D}$-$F^{Text}$ loss, respectively (refer to Section 3.4). (**b**) Testing. We use cosine similarity loss between per-point features and text features to perform open-vocabulary 3D Scene Understanding tasks. The ‘*an XX in a scene*’ serves as the input prompt for text, where ‘*XX*’ represents query text, which adopts a dataset class during the segmentation task.

**Figure 4 sensors-24-06166-f004:**
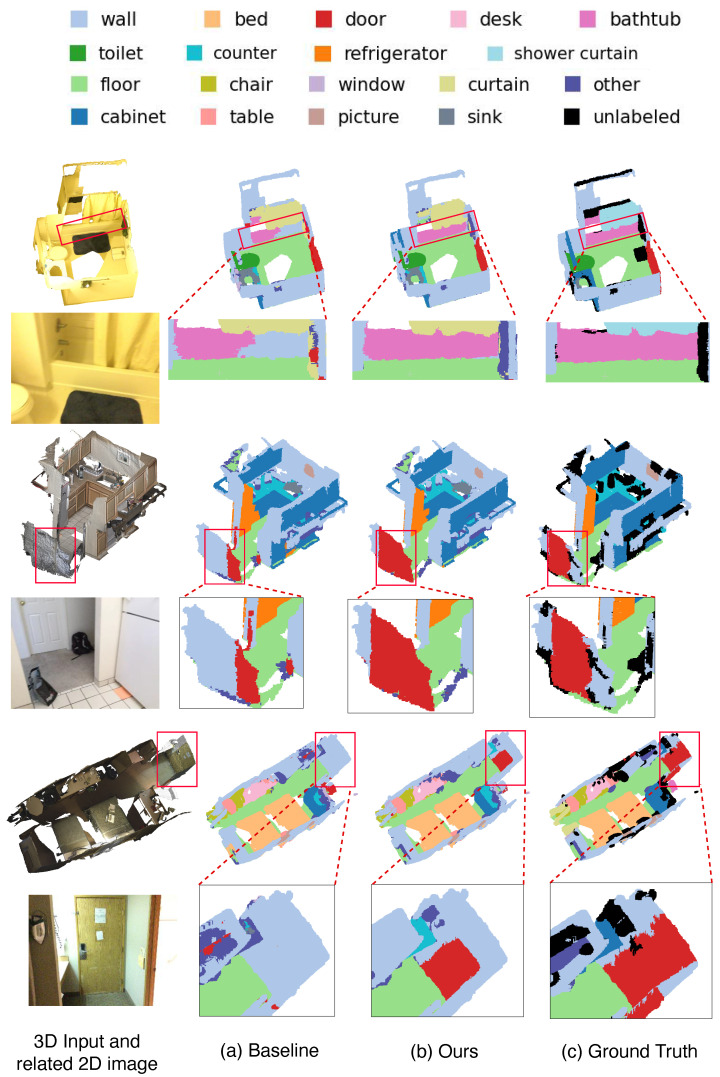
Qualitative results on ScanNet [26]. From left to right: 3D input and related 2D image, (**a**) the result of the baseline method (OpenScene [7]), (**b**) the proposed method, (**c**) the ground truth segementation.

**Table 1 sensors-24-06166-t001:** Comparisons between TAMC and other SOTA methods on ScanNet [26]. † means results reproduced by us.

Method	mIoU	mAcc
*Fully supervised methods*		
TangentConv [27]	40.9	-
TextureNet [29]	54.8	-
ScanComplete [30]	56.6	-
DCM-Net [31]	65.8	-
Mix3D [28]	**73.6**	-
VMNet [32]	73.2	-
MinkowskiNet [15]	69.0	77.5
*Zero-shot methods*		
MSeg-Voting [33]	45.6	54.4
OpenScene † [7]	50.6	62.2
TAMC	**51.9**	**63.4**

**Table 2 sensors-24-06166-t002:** mIoU scores for various loss weights (α in Equation (Equation 10)) applied to Textual Alignment. For these ablation studies, we utilize the full 3D features to eliminate the effect of Masked Consistency.

Loss Weight	mIoU
baseline(0)	50.6
0.001	50.6
0.005	51.0
0.01	50.4
0.05	**51.1**
0.1	49.5
0.5	30.1
1.0	16.5

**Table 3 sensors-24-06166-t003:** mIoU scores of different masking rations. MC means the performance of masked multimodal consistency training; MC → TA means we first train the model w/MC and then fine-tune it with the text-3D alignment loss; TA → MC means we first train the model w/text-3D alignment loss and then fine-tune it with the Masked Consistency training policy.

Masking Ratio	MC	MC → TA	TA → MC
0.1	50.6	50.0	51.3
0.3	50.8	49.7	51.5
0.6	51.4	49.7	51.4
0.9	51.5	50.5	51.7
0.95	**51.6**	50.1	**51.9**
0.99	51.3	**51.0**	51.1

**Table 4 sensors-24-06166-t004:** Per-class statistical comparison of open-world 3D semantic segmentation on ScanNet in terms of the IoU score. The proposed TAMC achieves higher accuracy than the baseline model across most classes.

Method	Wall	Floor	Cabinet	Bed	Chair	Sofa	Table	Door	Window	Bookshelf	Picture	Counter	Desk	Curtain	Fridge	Shower Curtain	Toilet	Sink	Bathtub	Other Furniture
Baseline [5]	72.8	86.6	43.4	70.3	67.5	**65.3**	52.5	43.4	49.0	63.7	**19.9**	34.6	45.0	52.0	39.4	0.0	**77.6**	49.6	57.6	21.9
TAMC	**73.9**	**88.7**	**44.6**	**71.2**	**71.3**	63.5	**53.6**	**45.4**	**51.6**	**65.1**	17.1	**41.7**	**45.1**	**53.7**	**42.0**	0.0	75.1	**50.3**	**60.6**	**24.7**

## Data Availability

Publicly available datasets were analyzed in this study. This paper contains the links to the datasets.

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
