# Peer review of "TAMC: Textual Alignment and Masked Consistency for Open-Vocabulary 3D Scene Understanding"

_sensors, 2024, doi:10.3390/s24196166_

Round 1

Reviewer 1 Report

Comments and Suggestions for Authors

1.          Section 4.3.4 and Table 3: it is better to clearly address how different masking ratios were introduced into the dataset. It is highly suspicious that same masking ratio would result in different occlusions to the scene features (before segmentation) and affect the TA as well as the followed TA-MC.

2.          Similar to comment 1: Does masking ratio 0.99 mean only 1% of the scene data was presented?

3.          Similar to the above two comments, it seems to me the different ratios are not producing significant impact to the mIoU scores. The Bolded case ones were not significantly outperforming the others (the claimed better improvement was questionable).

4.          For the results presented in Figure 4, please address how the ground truth segmentation was obtained (supposed to be manually visual processing?). Also, except for the qualitative comparison in the red block area, can a quantitative or overall statistical comparison be made as well? If applicable, it is also preferred another column of RGB images provided aside for visual comparison.

5.          Results presented in Table 4 are better evidenced/supported with visual demonstrations like those in Figure 4. To be more scientifically solid, zoom-in view to those superior performed segmentations would help convincing readers the quality of the proposed approach.

Reviewer 2 Report

Comments and Suggestions for Authors

The Title: TAMC: Textual Alignment and Masked Consistency for Open-Vocabulary 3D Scene Understanding

In this work, a contrastive learning framework based on Textual Alignment and Masked Consistency Training for the open vocabulary 3D scene understanding task is proposed. The research is important in its field. I think that it can be accepted after addressing the following issues

The comments:

1-The Abstract is short and requires more detail about the methodology and results. In general, the abstract should make sense on its own, without reference to the actual research paper.

2- In the Abstract, the main contribution is not clear. What is the main problem statement that is addressed in this work. Please define it clearly in the Abstract.

Besides, the following sentences “training policy to make the model to understand the whole scene with only partial 3D features” should be grammatically correct.

3-The introduction section is weak and needs more recent information to be added to produce a scientific justification for the proposed work. In general, the structure of the introduction should have background information to understand the problem. Then the problem statement needs to be defined based on the current gaps to provides a motivation to propose the current work then how the resolution of the problem will be resolved.

Besides, why the proposed work is mentioned in Figure 1? It can be mentioned in the section of the methods in another way to express the methodology of the proposed work.

4- The section of related work must contain an overview of previous studies in the same field that are relevant to the current work to situate it within the existing body of knowledge and highlight the gaps or limitations that aim to be addressed. The key findings and methodologies from previous works must be summarized to establish the context and importance of this work to prior work. More existing should be added and discussed.

5-There are various grammatical errors throughout the manuscript that need to be dealt with. Furthermore, the word “our” is repeated many times. It is preferable to replace it with other words that correspond to it.

6- The methods and results sections are good but need to be more organized in terms of presenting context.

7- The conclusion section should restate the proposed work and summarize the important points of the results to remind the reader of the importance of the work presented in the paper. Some future directions may also be added.

Comments on the Quality of English Language

Moderate editing of English language required.

Round 2

Reviewer 2 Report

Comments and Suggestions for Authors

The authors have addressed the comments correctly in the revised version of the manuscript and I have no more comments. It would just be better to replace the word "believe" in the Abstract with a more powerful word.

Comments on the Quality of English Language

 Minor editing of English language required.